# In Vivo Lentiviral Gene Delivery of HLA-DR and Vaccination of Humanized Mice for Improving the Human T and B Cell Immune Reconstitution

**DOI:** 10.3390/biomedicines9080961

**Published:** 2021-08-05

**Authors:** Suresh Kumar, Johannes Koenig, Andreas Schneider, Fredrik Wermeling, Sanjaykumar Boddul, Sebastian J. Theobald, Miriam Vollmer, Doreen Kloos, Nico Lachmann, Frank Klawonn, Stefan Lienenklaus, Steven R. Talbot, André Bleich, Nadine Wenzel, Constantin von Kaisenberg, James Keck, Renata Stripecke

**Affiliations:** 1Laboratory of Regenerative Immune Therapies Applied, REBIRTH-Research Center for Translational Regenerative Medicine, D-30625 Hannover, Germany; kumar.suresh@mh-hannover.de (S.K.); koenig.johannes@mh-hannover.de (J.K.); schneider.andreas@mh-hannover.de (A.S.); m.vollmer95@web.de (M.V.); 2Department of Hematology, Hemostasis, Oncology and Stem Cell Transplantation, Hannover Medical School, D-30625 Hannover, Germany; 3German Centre for Infection Research (DZIF), DZIF Partner Site Hannover-Braunschweig, D-30625 Hannover, Germany; 4Division of Rheumatology, Department of Medicine Solna, Center for Molecular Medicine, Karolinska University Hospital and Karolinska Institute, 17177 Solna, Sweden; fredrik.wermeling@ki.se (F.W.); sanjaykumar.boddul@ki.se (S.B.); 5Department of Internal Medicine I, Faculty of Medicine and University Hospital of Cologne, University of Cologne, D-50924 Cologne, Germany; sebastian.theobald@uk-koeln.de; 6Center for Molecular Medicine Cologne (CMMC), Faculty of Medicine and University Hospital of Cologne, University of Cologne, D-50924 Cologne, Germany; 7Institute of Experimental Hematology, Hannover Medical School, D-30625 Hannover, Germany; kloos.doreen@mh-hannover.de; 8Department of Pediatric Pneumology, Allergology and Neonatology, Hannover Medical School, D-30625 Hannover, Germany; lachmann.nico@mh-hannover.de; 9Biostatistics Group, Helmholtz Centre for Infection Research, D-38124 Braunschweig, Germany; frank.klawonn@helmholtz-hzi.de; 10Institute for Information Engineering, Ostfalia University, D-38302 Wolfenbuettel, Germany; 11Institute for Laboratory Animal Science, Hannover Medical School, D-30625 Hannover, Germany; Lienenklaus.Stefan@mh-hannover.de (S.L.); Talbot.Steven@mh-hannover.de (S.R.T.); bleich.andre@mh-hannover.de (A.B.); 12Institute for Transfusion Medicine and Transplant Engineering, Hannover Medical School, D-30625 Hannover, Germany; wenzel.nadine@mh-hannover.de; 13Department of Obstetrics, Gynecology and Reproductive Medicine, Hannover Medical School, D-30625 Hannover, Germany; vonKaisenberg.Constantin@mh-hannover.de; 14The Jackson Laboratory, Sacramento, CA 95838, USA; james.keck@jax.org

**Keywords:** humanized mice, stem cell transplantation, HLA match, lentiviral vector, vaccine, B cell maturation, class-switch, IgG, cytomegalovirus

## Abstract

Humanized mouse models generated with human hematopoietic stem cells (HSCs) and reconstituting the human immune system (HIS-mice) are invigorating preclinical testing of vaccines and immunotherapies. We have recently shown that human engineered dendritic cells boosted *bonafide* human T and B cell maturation and antigen-specific responses in HIS-mice. Here, we evaluated a cell-free system based on in vivo co-delivery of lentiviral vectors (LVs) for expression of a human leukocyte antigen (HLA-DRA*01/ HLA-DRB1*0401 functional complex, “DR4”), and a LV vaccine expressing human cytokines (GM-CSF and IFN-α) and a human cytomegalovirus gB antigen (HCMV-gB). Humanized NOD/Rag1^null^/IL2Rγ^null^ (NRG) mice injected by i.v. with LV-DR4/fLuc showed long-lasting (up to 20 weeks) vector distribution and expression in the spleen and liver. In vivo administration of the LV vaccine after LV-DR4/fLuc delivery boosted the cellularity of lymph nodes, promoted maturation of terminal effector CD4^+^ T cells, and promoted significantly higher development of IgG^+^ and IgA^+^ B cells. This modular lentigenic system opens several perspectives for basic human immunology research and preclinical utilization of LVs to deliver HLAs into HIS-mice.

## 1. Introduction

Human immune system (HIS)-mice are generated with human stem cell transplantation (HCT) applied to immunodeficient mouse strains. This type of model is a practical option to investigate human immunobiology and test human-specific biomedicines without the ethical concerns of research performed on human subjects. From the inception of primordial HIS-models in the 80′s until today, considerable progress has been achieved. The development of advanced severely immunodeficient mouse strains containing multiple mutations or transgenesis allowed for high engraftment of human HSCs and enduring long-term human immune reconstitutions [1,2]. Shultz, Ishikawa, and colleagues pioneered the transplantation of CD34^+^ hematopoietic stem cells (HSCs) into immunodeficient mouse strains lacking the common interleukin-2 receptor gamma chain (IL2Rγ) (NOD/Rag1^null^/IL2Rγ^null^–NRG mice, NOD/LtSz-scid/IL2Rγ^null^–NSG mice) [3]. The NSG strain and its derivatives are broadly used, and they fully lack murine T, B, and NK cells. In addition, different human grafts were used to transplant NSG mice: purified CD34^+^ HSCs derived from human fetal liver or umbilical cord blood (CB), combining fetal bone marrow (BM), liver, and thymus tissues (BLT-mice), or additional lung tissue (BLT-L-mice) [4]. These models have been extensively explored to study human infections, cancer, and metabolism [5].

After HSC engraftment in the mouse BM, human progenitor cells reach the mouse thymus and liver. This is followed by a selection of naïve T cells restricted by the T cell receptor (TCR) to antigens presented by the mouse class I and II major histocompatibility complexes (MHCs). The naïve T cells subsequently biodistribute in secondary lymphatic tissues, where they can be primed by professional antigen-presenting cells such as dendritic cells (DCs) to generate memory and effector T cells. DCs play a central role in lymphatic tissues that are key for immune synapses with T and B cells and for stimulation of specific and long-lasting immunity [6,7]. Yet, some key limitations of HIS-mouse models that are currently being addressed are: (i) lack of matching of the major histocompatibility complexes (MHCs)/human leukocyte antigens (HLAs) between the mouse and human cells, (ii) low levels of circulating human cytokines and growth factors capable to stimulate human cells, and (iii) underdeveloped lymph nodes. These issues remain to be resolved for effective priming and boosting of human T and B cells to mount functional cellular and humoral responses, respectively.

One strategy to improve the human adaptive immunity explored so far is the utilization of transgenic mouse strains that match a few HLAs between mouse tissues and human cells. Of particular relevance is the expression of the HLA class II (HLA-II) dimers, which mediate the interaction of professional antigen-presenting cells (APCs) with CD4^+^ T cells. CD4^+^ T cells are involved in directing the cytokine milieu toward reactivity or tolerance and participate in CD4-dependent B cell development [8]. The main challenge for HLA matching between donors and hosts is that HLA class II molecules are largely polymorphic. The HLA-DRB1*04:01 (shortened to “DR4”) is a relatively frequent HLA class II haplotype found in Caucasians (19–65%), Hispanics (17–26%), and African descendants (4–40%) [9]. NRG or NOD-Shi/scid/ IL2Rγc^−/−^ (NOG) mice were engineered to express DR4 and HCT with DR4^+^ CB-derived HSCs, and demonstrated a significantly enhanced development of mature T cells, accompanied by increased antibody class switching in B cells and production of human IgGs [10,11]. The DR4^+^ NRG (DRAG) humanized mice have been successfully used in the infectious disease field, demonstrating human humoral responses against different pathogens [12,13]. Nonetheless, the use of transgenic strains is limited so far to a few HLA-DR haplotypes. Taking into account the fact that the HLA-II haplotypes are highly polymorphic, several thousands of different transgenic strains would have to be produced to cover all the human HLA-II repertoire.

Previously, we have explored NRG mice transplanted with human CD34^+^ HSCs and immunized with autologous-induced DCs (iDCs) expressing high levels of HSC-matched HLAs. For the generation of iDCs, isolated monocytes were transduced with tricistronic lentiviral vectors (LVs) co-expressing human granulocyte-macrophage colony-stimulating factor (GM-CSF) and human interferon-α-2b (IFN-α), and were therefore internally loaded with full-length antigens [14,15,16,17]. Remarkably, administration of iDCs shortly after HCT profoundly improved the development of lymph node-like structures in HIS-mice, including the presence of follicular T helper (FTh) cells, maturation of antigen-specific terminal effector (TE) T cells, and IgG^+^ B cells [14,15,16,17]. iDCs are currently in clinical development as an adoptive cell therapy to accelerate the reconstitution of the adaptive immunity of immune-compromised patients after transplantation, in order to protect them against human cytomegalovirus (HCMV) reactivation and leukemia relapse [17,18,19]. For research utilization, however, the production of iDCs is technically and logistically demanding.

Therefore, we considered exploring the use of LVs to improve the adaptive immune reconstitution of HIS-mice since they provide a safe, robust, persistent, non-toxic, and low immunogenicity vector platform [20]. LVs infect a broad range of cells and can be applied directly in vivo, for example, for vaccination [21]. LVs can also be used for in vivo engineering of T cells expressing chimeric antigen receptors [22]. We had shown in previous work that LVs containing the CMV promoter promoted high expression of different transgenes and inflammatory responses when applied intravenously (i.v.) into C57BL/6 mice [23]. LVs with the CMV promoter driving expression of tyrosinase-related protein 2 (TRP2) applied as a vaccine i.v. effectively protected C57BL/6 mice against a B16 melanoma challenge [23]. In the same study, we also tested LVs containing the HLA-DR-alpha minimal promoter for expression of the firefly luciferase (fLuc) or green fluorescent protein (GFP) reporter genes. When LV-DRp-fLuc was injected i.v. into nude and C57BL/6 mice, persistent fLuc expression was observed for up to 39 weeks by bioluminescence imaging (BLI) analyses, mostly in spleen and liver tissues [23]. LV-DRp-GFP applied i.v. into C57BL/6 mice transduced specifically MHCII^+^ cells in the marginal zone of spleen and non-parenchymal cells of liver [23].

Here, a tricistronic vector containing the DR promoter was constructed for co-expression of the HLA-DR4 dimer and fLuc. The vector effectively resulted in DR4 expression in several transduced cell lines and primary cells. Cells transduced for DR4 expression and pulsed with peptides stimulated DR4-restricted T cells in vitro. Administration of LV-DR4/fLuc vector into humanized NRG (huNRG) mice resulted in persistent fLuc expression in the spleen and liver. We then combined i.v. administration of LV-DR4/fLuc with a LV vaccine expressing GM-CSF, IFN-α, and the HCMV glycoprotein B gB into huNRG mice. The combination was safe, showed no toxicity, and enhanced the cellularity of lymph nodes compared with non-treated or single-treated controls. A correlation trend was observed between the combined LV treatments and the development of terminal effector CD4^+^ T cells. Importantly, the LV combination significantly enhanced the development of IgG^+^ and IgA^+^ B cells in the spleen. This is the first proof-of-concept exploring in vivo systemic LV-mediated HLA-DR delivery and vaccination to promote the development of human class-switched B cells in HIS-mice.

## 2. Materials and Methods

### 2.1. Cell Lines

NIH/3T3 (mouse embryonic fibroblasts; from now on called 3T3, ATCC, Manassas, VA, USA, kindly provided by Dr. Constanca Figueiredo, MHH, Hannover) and HEK293T cells (human embryonic kidney cells, from now on called 293T, ATCC) were cultured in DMEM (ThermoFisher, Waltham, MA, USA) containing 10% fetal bovine serum (FBS, HyClone, Logan, UT, USA) and 1% penicillin/streptomycin (Merck Millipore, Billerica, MA, USA) at 37 °C with 5% CO_2_. K562 cells (human myelogenous leukemia cell line, ATCC) were cultured in RPMI (ThermoFisher) containing 10% FBS and 1% penicillin/streptomycin at 37 °C with 5% CO_2_. Kasumi-1 cells (human myeloid leukemia cell line, ATCC) were cultured in RPMI (ThermoFisher) containing 20% FBS and 1% penicillin/streptomycin at 37 °C with 5% CO_2_. HL-60 cells (human leukemia cell line, ATCC, kindly provided by Dr. Michael Heuser, MHH, Hannover) were cultured in IMDM (STEMCELL Technologies, Vancouver, BC, Canada) containing 20% fetal bovine serum and 1% penicillin/streptomycin at 37 °C with 5% CO_2_.

### 2.2. Primary Cells

Umbilical cord blood (CB) samples were collected after informed consent from the mothers at the Department of Gynecology and Obstetrics (MHH, Hannover) and obtained according to study protocols approved by MHH Ethics Review Board (approval number 4837 to RS). All CB samples used here were genotyped to determine the HLA alleles and only DR4-negative CBs were used. Mononuclear cells (MNCs) from CB were isolated by Ficoll gradient centrifugation as described previously [16]. Shortly, CD34^+^ hematopoietic stem cells (HSCs) were isolated after two rounds of positive selection with MACS magnetic beads according to the manufacturer’s instructions (CD34 MicroBead Kit; Miltenyi Biotec, Bergisch Gladbach, Germany). For transduction, CB CD34^+^ cells were cultured in X-VIVO 15 medium (Lonza, Verviers, Belgium) containing 1% penicillin/streptomycin and human growth factors (stem cell factor, SCF; Flt3 receptor ligand, Flt3L; Thrombopoietin, TPO; all as 100 ng/mL; R&D Systems, Minneapolis, MN, USA) at 37 °C with 5% CO_2_. For transduction of murine HSCs recovered from mice, Lineage-negative (Lin^−^) cells isolated from BM were cultured in StemSpan™ SFEM (STEMCELL Technologies) containing 1% penicillin/streptomycin and mouse growth factors (SCF (10 ng/mL); TPO (20 ng/mL); fibroblast growth factor, FGF (10 ng/mL); and insulin-like growth factor, IGF (20 ng/mL), all from PeproTech, London, UK at 37 °C with 5% CO_2_.

### 2.3. Construction, Production, and Titering of Lentiviral Vectors

For all our studies, we used self-inactivating (SIN) LV backbones with reduced risks to cause insertional mutagenesis [24,25]. Cloning was performed using sequence- and ligation-independent cloning (SLIC) techniques adapted from Li and Elledge [26]. Briefly, inserts were amplified by PCR using designed primers with approximately 30 base pairs homologous to adjacent sequences in the final construct. Following T4 DNA polymerase (New England Biolabs, Ipswich, MA, USA) treatment, 5′-overhangs were generated. These overhangs allowed annealing between the recipient plasmid vectors and the inserts, and the mixture was used to transform chemically competent bacteria (produced in-house, derived from XL-10 Gold Ultracompetent cells, Stratagene, San Diego, CA, USA). Upon transformation, homologous DNA recombination in bacteria resulted in the final plasmid constructs. 2A elements were included in the primer sequences to allow the generation of multicistronic cassettes. LV-G2α/gB contains a CMV promoter driving simultaneous expression of human GM-CSF, IFN-α, and HCMV-gB. The transgenes are interspaced with two non-homologous sequences derived from the porcine teschovirus-1 2A element (P2A). This construct was mentioned in previous work for the generation of iDCgB [17]. The LV-DR4/fLuc vector contains a minimal HLA-DR-alpha promoter [23] and encodes the beta chain HLA-DRB1*04:01:01:01 (266 aa, 798 bp), the alpha chain HLA-DRA1*01:01 (254 aa, 762 bp sequence), and firefly luciferase (fLuc). The transgenes are interspaced with a thosea asigna virus 2A element (T2A) and with a foot-and-mouth disease virus 2A element (F2A). The structures of the vectors were confirmed by restriction digestions and the multicistronic sequences were confirmed by sequencing analyses. LV particles were produced as a third-generation packaging system in transiently transfected 293T cells. Transfection of the transfer plasmid and three packaging plasmids (pMD.G expressing VSV-G; pRSV/Rev expressing Rev; pMDLg/pRRE expressing Gag, Pol, and RRE) was performed as previously reported [19,27]. Briefly, 293T cells were seeded into culture flasks (Sarstedt, Nümbrecht, Germany), and transfection was performed on the next day using polyethylenimine (PEI, Polysciences, Warrington, PA, USA). The medium was changed 24 h after transfection. Cell supernatants were harvested 24 h and 48 h after the media exchange, filtered through 0.45 µm (Sarstedt, Nümbrecht, Germany), and 24 h and 48 h LV supernatants were pooled and concentrated twice by high-speed centrifugation. The virus pellets were resuspended in PBS and stored at −80 °C. The concentration of viral particles was determined as p24 equivalents/mL using anti-p24 HIV ELISA according to the manufacturer’s instructions (QuickTiter™ Lentivirus Titer Kit, Cell Biolabs, San Diego, CA, USA).

### 2.4. Confirmation of Transgene Expression

1 × 10^5^ 293T cells were transduced with 250 ng of p24 equivalents of the LV-G2α/gB vector in 1 mL of culture in the presence of protamine sulfate (5 µg/mL; Valeant, Düsseldorf, Germany) and expanded for one passage. For detection of secreted GM-CSF and IFN-α, the cells were seeded at a density of 10^6^ cells/mL in a 6-well-plate. 48 h after seeding, the supernatants were harvested and stored at −80 °C. Detection of GM-CSF and IFN-α was performed by ELISA according to the manufacturer’s instructions (Human GM-CSF ELISA development kit and Human IFN-α subtype 2 ELISA development kit, MABTECH AB, Nacka Strand, Sweden). For detection of HCMV-gB, 1 × 10^5^ 293T cells were transduced with 350 ng of p24 equivalents of the LV-G2α/gB vector as described above and expanded for one passage. Cells were detached, stained against gB using mAb p27-287 plus secondary AF488-conjugated anti-mouse-IgG (Appendix A), and analyzed by flow cytometry using an LSR II cytometer (BD Biosciences, San Jose, CA, USA). For detection of the DR4 dimer, 1 × 10^5^ cells (3T3, Kasumi, K562 or HL-60 cells) were transduced with 1 µg of p24 equivalents of LV-DR4/fLuc as described above. 2 × 10^5^ CB CD34^+^ cells were transduced with 1 µg of p24 equivalents of LV-DR4/fLuc vector in 200 µL of culture in the presence of poloxamer 407 (100 µg/mL; Merck, Darmstadt, Germany). 3 × 10^5^ mouse BM Lin^−^ cells were transduced with 3 µg of p24 equivalents of LV-DR4/fLuc vector in 1 mL of culture in the presence of olybrene (64 ng/mL; Merck, Darmstadt, Germany). Four days post-transduction, cells were harvested and immune stained with the HLA-DR antibody (clone L243, BioLegend, San Diego, CA, USA) for detection of the DR4 dimer (Appendix A) and analyzed by LSR II cytometer (BD Biosciences, San Jose, CA, USA). For analyses of luciferase reporter expression, 3T3 cells were harvested 4 days post-transduction, washed with PBS, and lysed using 20 µL lysis reagent according to the manufacturer’s instructions (Luciferase Assay Systems kit; Promega, Madison, WI, USA), and 20 μL of cell lysate mixed with 100 μL of luciferase assay reagent and bioluminescence was measured using a Tristar^2^ Microplate Reader (Berthold Technologies, Bad Wildbach, Germany).

### 2.5. In Vitro Activation of 58-T Cells with DR4^+^ 3T3 Cells Generated with LV-DR4/fLuc

The mouse-hybridoma 58-T cell lines expressing TCRs restricted to DR4 epitopes of glutamic acid decarboxylase (GAD65) and H3N2-Influenza hemagglutinin (HA) and expressing an NFAT-GFP reporter system, hCD4, and the fluorochrome Ametrine were generated as described [28]. 1 × 10^4^ 3T3 cells (Mock or DR4^+^) were seeded in 200 μL DMEM medium per well in U-bottom 96-well plates. After overnight incubation, 58-T cells (targeted against Influenza H3N2 Hemagglutinin (HA) or glutamate decarboxylase (GAD65)) were added at 2 × 10^4^ cells/well and DR4-binding peptides were added (HA_306–318_: PKYVKQNTLKLAT; GAD_115–127_: MNILLQYVVKSFD; synthesized by GenScript, Piscataway Township, NJ, USA; stock solution 10 mg/mL in DMSO). As a control for 58-T cell activation, anti-CD3/CD28 (BioLegend) was used. Following 24 h incubation, cell culture supernatants were collected for IL-2 ELISA (BioLegend, San Diego, CA, USA). Cells were analyzed for NFAT-GFP expression by flow cytometry using a BD B6 Accuri or a BD FACSVerse (BD Biosciences), and data were analyzed with FlowJo V10 software (Treestar, Ashland, OR, USA). The NFAT-GFP expression was assessed in the 58-T cells by gating on viable singlets, and further on FSC^high^ and/or Ametrine^+^ cells to distinguish the 58-T cells in the co-culture.

### 2.6. Generation of Humanized NRG Mice

The animal protocols for mouse studies were approved by the Lower Saxony Office for Consumer Protection and Food Safety–LAVES (approval Nr. 33.19-42502-04-16/2222 and 33.19-42502-04-19/3336) and performed according to the German animal welfare act and the EU directive 2010/63. Breeding pairs of NRG mice (stock number 017914, *NOD.Cg-Rag1tm1Mom IL-2Rγctm1Wjl/SzJ* [29]) and DRAG mice (stock number 017914, *NOD.Cg-Rag1^tm1Mom^Il2r**γ**^tm1Wjl^**Tg(HLA-DRA, HLA-DRB1*0401)39-2Kito/ScasJ* [10]) were obtained from The Jackson Laboratory (JAX; Bar Harbor, ME, USA) and bred and maintained in-house under pathogen-free conditions. Tissues of DRAG mice were used as controls to evaluate DR4 expression in mouse HSCs. HCT was performed as described [16], and we used both males and females as recipients. Briefly, 5–6 weeks old mice were sub-lethally irradiated (450 cGy) using a [^137^Cs] column irradiator (Gammacell 3000 Elan; Best Theratronics, Ottawa, ON, Canada), and 4 h after irradiation, 2.0 × 10^5^ human CB-CD34^+^ cells were injected i.v. into the tail vein of mice. CB CD34^+^ units used in the studies were pre-tested before experiments. Only CB units that resulted in long-term human reconstitution (>20% huCD45^+^ cells in peripheral blood from 10–15 weeks after HCT) were used for experiments. The body weight and general health of the mice were monitored weekly after HCT.

### 2.7. In Vivo Administration of LVs into Mice and Longitudinal BLI Analyses

Next, 1–3 µg of p24 equivalents of LV-DR4/fLuc were injected i.v. into the tail vein of NRG or humanized (hu)NRG mice (1-week post-HCT). For immunization, 1 µg of p24 equivalents of LV-G2α/gB vaccine (VAC) was injected as a prime-boost i.v. into huNRG mice at 6 and 7-weeks post-HCT. To visualize fLuc expression, mice were analyzed at several intervals by BLI analyses using the IVIS SpectrumCT (PerkinElmer, Waltham, MA, USA) as described [30]. Briefly, mice were anesthetized using isoflurane and shaved. Five minutes before imaging, mice were injected intraperitoneally (i.p.) with 2.5 µg D-Luciferin potassium salt (SYNCHEM, Elk Grove Village, IL, USA), freshly reconstituted in 100 µL PBS, and optical imaging analyses were performed with IVIS SpectrumCT. Pictures were acquired in a field of view C, f stop 1, and medium binning for each mouse. Exposure time was kept to 300 s for each mouse. Data were analyzed using LivingImage software (PerkinElmer, Waltham, MA, USA). The anatomical regions of interest (ROI) were kept constant for quantified analyses of all mice, and photons/second (p/s) were calculated.

### 2.8. Flow Cytometry Analyses of Human T and B Cells

Immune-reconstitution of human CD45/CD3/CD19/CD4/CD8 positive cells in the peripheral blood lymphocytes (PBL) was monitored at 10, 15, and 20 weeks after HCT. At the endpoint analyses, PBL was collected and several tissues were biopsied (spleen, bone marrow, and lymph nodes) and processed as previously reported [17]. Spleen and PBL samples were incubated with lysis buffer (0.83% ammonium chloride/20 mM HEPES, pH 7.2) for 5 min at room temperature to remove erythrocytes. Tissues were processed as single-cell suspensions, blocked in PBS plus 10% FBS, stained with optimum concentration of antibodies for flow cytometry (Appendix A), and additional washing was performed to remove unbound antibodies. For data acquisition, an LSR II flow cytometer (BD Biosciences) was used and analysis was performed using FlowJo software. Surplus samples were cryopreserved in cryomedium (40% PBS; 50% Human Serum, Sigma-Aldrich, St. Louis, MO, USA; and 10% DMSO) and stored at −150 °C for further analysis. The gating strategy for analyses of human T and B cells from PBL is shown in Appendix A. Exemplary gating for analyses of T and B cells obtained from the spleen is shown in Appendix A.

### 2.9. ELISA Analyses for Detection of Anti-gB IgG in Mouse Plasma

Next, 96-well plates (Corning, Corning, NY, USA) were coated with 0.05 M sodium carbonate buffer containing 2 µg of purified recombinant gB protein (kindly provided by Dr. Thomas Krey, University of Lübeck) at 4 °C overnight. Washing was performed five times using PBS plus 0.1% Tween 20 (Bio-Rad, Hercules, CA, USA), followed by blocking with PBS plus 10% FBS for 2 h at 37 °C. After repeated washing, plasma from mice was added and incubated for 2 h at 37 °C. Dilutions were performed in PBS plus 2% FBS. For staining with a secondary antibody (Appendix A), the plate was washed again five times and the samples were incubated for 45 min at 37 °C with anti-human IgG conjugated with horseradish peroxidase (HRP, Bio-Rad) diluted at 1:5000 in PBS plus 2% FBS. Plates were washed again, and 3,3′,5,5′-Tetramethylbenzidin (TMB) (ThermoFisher) solution was added. After approximately 15 min, the reaction was stopped with 100 μL H_2_SO_4_ (Roth, Karlsruhe, Germany) before the OD450 was acquired with an ELISA plate reader (Hidex, Mainz, Germany).

### 2.10. Statistical Analyses

For analyses of data obtained in vitro, *t*-tests or two-way ANOVA with Sidak’s multiple comparison tests were used. For in vivo data, the two-sided, unpaired Welch *t*-test was applied for pairwise comparison of values in two groups. For skewed data—especially for those depicted in log-scale—the logarithm was taken before applying the *t*-test. Means and sample standard deviations were computed based on the original scale of the data without taking the logarithm. The significance level was chosen as 0.05. Statistical analyses were carried out using GraphPad Prism V7.0 software (GraphPad Software, La Jolla, CA, USA) and R software V4.1.0 (R Foundation for Statistical Computing, Vienna, Austria).

## 3. Results

### 3.1. Lentiviral Vector Designs and Titer

Two tricistronic vectors were generated: LV-G2α/gB and LV-DR4/fLuc. The LV-G2α/gB was used as a vaccine (VAC) to induce immune responses against HCMV-gB exposed as a trimer on the cell surface and co-adjuvanted by expression of secreted GM-CSF and IFN-α (Figure 1A). The LV-DR4/fLuc vector was used for HLA-DR4 gene delivery and was designed to express the DR4 dimer on the cell surface and intracellular firefly luciferase (fLuc) for tracking the vector expression by BLI analyses (Figure 1B). High titers of the concentrated vectors were obtained, in the ranges of 15 μg/mL HIV-p24 equivalents for LV-G2α/gB and 22 μg/mL HIV-p24 equivalents for LV-DR4/fLuc (Figure 1C). In our laboratory, 1 μg of HIV-p24 equivalents correspond to about 1 × 10^7^ infective LV particles measured by quantitative real-time PCR analyses [19].

### 3.2. Detection of Transgenes in Transduced 293T and 3T3 Cells

The LV-G2α/gB vector was validated in ex vivo transduced 293T cells. Secreted GM-CSF (250 ng/mL) and IFN-α (1200 ng/mL) were detected by ELISA in supernatants from 293T cells transduced with LV-G2α/gB (Figure 2A), and >95% of the transduced cells expressed HCMV-gB on the cell surface (Figure 2B). The LV-DR4/fLuc vector was tested in transduced mouse 3T3 fibroblasts. The assembly of the HLA-DR dimer on the cell surface was confirmed by immune staining with a fluorochrome-conjugated monoclonal antibody (clone L243) recognizing the HLA-DR dimer (Figure 2C). Next, 3T3 cells transduced with LV-DR4/fLuc were analyzed by luminometry. Detection of the luminescent signal was correlated with the numbers of cells used in the assay, and the detection limit was approximately 200 cells per well (Figure 2D).

### 3.3. Mouse and Human Hematopoietic Cells Express DR4 after Transduction with LV-DR4/fLuc

Next, we examined the expression of DR4 on the surface of the mouse and human hematopoietic cells transduced with LV-DR4/fLuc. BM of NRG mice was used to prepare lineage negative (Lin^−^) HSCs. Three days post transduction, flow cytometry analyses of HLA-DR showed a transduction efficiency of around 20%, whereas Lin^−^ HSCs obtained from DRAG mice showed 30% of the cells expressing detectable HLA-DR (Figure 3A). Transduction of the human myeloid leukemia cell lines Kasumi, HL-60, and K562 resulted in approximately 50–80% HLA-DR^+^ cells (Figure 3B). Transduction of primary CB CD34^+^ cells showed a significant upregulation in the levels of HLA-DR (*p* = 0.0231, Figure 3C). Taken together, transduction of different types of hematopoietic cells with LV-DR4/fLuc resulted in consistent DR4 gene transfer and expression on the cell surface.

### 3.4. Functionality of Transgenic DR4 to Present Peptides to DR4-Restricted T Cells

A functional assessment of DR4 was performed by co-culture of LV-DR4/fLuc-transduced 3T3 mouse fibroblasts and two different TCR^+^ CD4^+^ 58-T cells clones restricted to DR4- described in previous work [28]. As a reporter cell system, 58-T cells contain the GFP transgene downstream of the NFAT-promoter, such that signaling through the TCR leading to transactivation of the NFAT promoter leads to GFP expression, which is detectable by flow cytometry (see Appendix A for FACS gating strategy). In this experiment, we tested if DR4^+^ 3T3 cells could function as artificial APCs presenting specific antigenic peptides. DR4^+^ 3T3 cells were loaded with either HAor GAD peptides and cocultured with the cognate antigen-restricted 58-T cells. As a positive control, 58-T cells were non-specifically activated with anti-CD3/CD28 antibodies. Co-culture of DR4^+^ 3T3 cells pulsed with GAD65 peptide and the corresponding anti-GAD65 TCR^+^ 58-T cells led to significantly (*p* < 0.0001) higher activation of the 58-T cells than Mock 3T3 cells or when non-specific peptides were used in the assay. T cell activation was directly correlated with the peptide levels (Figure 4A). These results were confirmed in a parallel experiment, utilizing DR4^+^ 3T3 cells pulsed with HA peptide and anti-HA TCR^+^ 58-T cells as reporters (Figure 4B). Additionally, we measured the concentration of IL-2 secreted by activated 58-T cells in the supernatants. Corroborating the NFAT-GFP reporter system, IL-2 secretion was significantly higher (*p* < 0.0001) when the antigen-specific 58-T cells were activated with peptide-pulsed DR4^+^ 3T3 cells than with Mock 3T3 cells, which again demonstrated T cell activation upon transduction and peptide loading (Figure 4D,E). These co-culture assays demonstrated in vitro functionality of the transgenic DR4 dimer to present peptides specifically to DR4-restricted T cells.

### 3.5. LV-DR4/fLuc Applied i.v. into NRG and huNRG Mice and In Vivo Transduction

We evaluated the feasibility of in vivo LV delivery into NRG and in huNRG mice. Three i.v. doses of LV-DR4/fLuc (0.3, 1.0, and 3.0 µg p24 LV equivalents) were tested in triplicate mice. The first set of analyses were performed with NRG mice, and no adverse effects were observed. For humanized NRG mice, LV was administered one week after HCT. For both types of experimental mice, the 0.3 μg dose resulted in variable results, and therefore this dose was discontinued (data not shown). Longitudinal BLI analyses were performed at several intervals up to 15 weeks after LV injection (Figure 5A), and 1.0 and 3.0 µg p24 LV equivalents applied i.v. showed a clear demarcation of vector expression in the anatomical regions of the liver, spleen, and occasionally bones, persisting over 15 weeks (Figure 5B,C, Appendix A). The quantified total Flux levels obtained for analyses of spleens indicated that the vector expression was comparable for NRG and huNRG. The 3.0 μg LV dose resulted in slightly higher bioluminescence levels than the 1.0 μg LV dose but for most time points the differences were not statistically significant.

### 3.6. LV-DR4/fLuc i.v. Delivery Does Not Lead to Acute or Chronic Graft-versus-Host Disease

HIS-mice can maintain the human immune reconstitution long-term. Sporadically, however, the mice can show an unbalance in the T cell central tolerance, and the human lymphocytes can be involved in acute or chronic xenogeneic graft-versus-host disease (GvHD) [31]. LV-DR4/fLuc administrations did not result in body weight loss (Figure 6A) and skin rashes, which are usual clinical signs of GvHD. Longitudinal analyses performed at 2, 7, and 12 weeks after LV administration also did not indicate any abrupt changes in the human immune reconstitution, such as high T cell frequencies in the peripheral blood (data not shown); 15–16 weeks after HCT is the time when the human T cells start showing increased expansion in the spleen of HIS-mice [17]. Flow cytometry analyses of CD4^+^ and CD8^+^ T cells in spleens at 15 weeks after vector administration showed that administration of 1.0 μg or 3.0 μg of p24 equivalents of LV-DR4/fLuc slightly increased the frequencies of CD4^+^ T cells when compared to control mice (Figure 6B) (same gating strategy as shown in Appendix A for peripheral blood, Appendix A). On the other hand, no differences were seen for the frequencies of CD8^+^ T cells. This is consistent with the hypothesis that higher HLA-DR expression would be particularly useful for the activation of CD4^+^ T cells. In summary, these results indicated that LV-DR4/fLuc i.v. administration into humanized mice promoted a slight increase in CD4^+^ T cell frequencies, but no GvHD was observed. For practical reasons regarding obtainability of the vector and to avoid inflammatory reactions and potential unwanted inflammatory responses, we opted for the 1 μg LV-DR4/fLuc dose for the subsequent experiments combining gene delivery of DR4 with vaccination.

### 3.7. Feasibility of LV-DR4/fLuc i.v. Application (DR4) Combined with LV-G2α/gB (VAC)

To further investigate if the LV-DR4/fLuc vector (DR4) administration after HCT could improve immunization effects in huNRG mice, we performed prime-boost vaccinations with LV-G2α/gB (VAC). One week after HCT, huNRG mice were administered i.v. with PBS or with 1 μg of LV-DR4/fLuc (Figure 7A). At weeks 6 and 7 after HCT, a cohort was then prime-boosted i.v. with 1 µg of VAC. Four groups were compared: PBS control (CTR, n = 6), VAC-only (VAC, n = 7), DR4-only (DR4, n = 11), and DR4 plus VAC (DR4/VAC, n = 9). The results correspond to pooled data obtained from four independent experiments performed with mice humanized with CD34^+^ cells isolated from four different CB donors (all of the donors negative for the *HLA-DRB1*04:01* allele). Analyses of BLI and the human immune reconstitution in blood were performed at several time points after HCT. At terminal endpoint 20–22 weeks after HCT, lymphatic tissues were collected and analyzed. Vaccination did not affect the expression of fLuc (Figure 7B), indicating that no induced immunogenicity was generated against the cells transduced with LV-DR4/fLuc. Although the fLuc signal was gradually decreased, it persisted up to 20–22 weeks and was comparable between DR4 and DR4/VAC (Figure 7C, Appendix A). Conclusively, the data showed that immunization with VAC did not affect the persistency of the LV-DR4/fLuc vector expression.

### 3.8. DR4 Combined with VAC Does Not Cause GvHD but Results in a Transient Increase in T Cell Frequencies

All mice were monitored weekly for body weight and clinical signs of GvHD until the terminal analyses to check whether combined administration of LV-DR4/fLuc and VAC would be associated with any signs of toxicity or GvHD. Within all cohorts, we did not find any evidence for toxicity or GvHD (Figure 8A). This result indicated that a combination of LV-DR4-fLuc and VAC administration at 1 μg each was feasible without any side effects. Next, we evaluated the combined effects of LV-DR4/fLuc and VAC on the human immune reconstitution. PBL samples were collected longitudinally and analyzed. Ten weeks after HCT (thus 4 weeks after the first vaccination), higher frequencies of CD4^+^ T cells (DR4/VAC versus control: 4.3 fold difference, *p* = 0.299) and CD8^+^ T cells (DR4/VAC versus control: 4.1 fold difference, *p* = 0.201) were observed (Figure 8B,C, Appendix A). This effect diminished gradually at weeks 15 and 20–22 post HCT. Thus, this data indicated that DR4 in combination with VAC primed early CD4^+^ and CD8^+^ T cell development in huNRG mice without causing GvHD.

### 3.9. DR4 Combined with VAC Promotes CD4^+^ and CD8^+^ T Cell Maturation

In the next step, we further analyzed CD4^+^ and CD8^+^ T cell memory responses in the spleen (SPL) > 14 weeks after the first immunization based on the expression of CD62L and CD45RA: naïve (N, CD62L^+,^ and CD45RA^+^), central memory (CM, CD62L^+,^ and CD45RA^−^), effector memory (EM, CD62L^−^ and CD45RA^−^), and terminal effector (TE, CD62L^−^ and CD45RA^+^) (see the gating strategy in Appendix A). After DR4/VAC combined administration, a reduction in the total numbers of EM CD4^+^ T cells in SPL was seen compared with CTR (Figure 9A), whereas a 2.4-fold relative increase in total numbers of TE CD4^+^ T cells was observed (Figure 9B, *p* = 0.068, Appendix A). For CD8^+^ T cells, DR4/VAC combination was associated with a 2.2-fold increase in total numbers of CD8^+^ EM (Figure 9C, *p* = 0.427, Appendix A) and a 1.6-fold increase in total numbers of TE cells (Figure 9D, *p* = 0.328, Appendix A) in the spleen of DR4/VAC group when compared with CTR. Thus, DR4/VAC combination in huNRG mice was associated with modest changes in the immunophenotype of human TE CD4^+^ and CD8^+^ T cells in the spleen, implying long-term in vivo effects on T cell activation.

### 3.10. DR4 Combined with VAC Promote B Cell Class Switch

Finally, we evaluated if the lentiviral administrations could improve the B cell development and maturation (see representative flow cytometry gating example in Appendix A). DR4/VAC combination significantly promoted an increase in the total number of class-switched B cells defined as CD19^+^IgM^−^IgG^+^ (Figure 10A, 4.71 fold-difference, *p* = 0.0125, Appendix A) and CD19^+^IgM^−^IgA^+^ (Figure 10B, 4.18 fold-difference, *p* = 0.0012, Appendix A) when compared with CTR group. VAC or DR4 alone did not produce these pronounced effects. The number of total CD19^+^IgM^+^ B cells within the spleen did not change significantly upon administration of DR4/VAC (Figure 10C). To conclude, we observed a correlation between the CD4^+^ T cell maturation trends promoted by DR4 /VAC and the absolute numbers of class-switched human IgG^+^ and IgA^+^ class-switched B cells.

### 3.11. VAC Promotes IgG Humoral Responses against gB

Ultimately, since B cell affinity maturation is known to occur in lymph nodes (LNs), we collected, pooled, and investigated cells recovered from peripheral and mesenteric LNs. We observed an increase in cellularity in LNs recovered from the DR4/VAC cohort compared with the control groups (Figure 11A, 2.06 fold-difference, *p* = 0.1375, Appendix A). To evaluate if the vaccinated mice could develop IgG^+^ humoral responses specific against the HCMV-gB antigen, plasma samples were analyzed by an ELISA assay using recombinant gB protein. Surprisingly, plasma obtained from mice in the VAC cohort showed detectable IgG^+^ reactivity against gB (Figure 11B, Appendix A). Two mice in the VAC cohort were clear immunization responders. As a reference to validate the assay, we used plasma samples from a human donor known to be HCMV positive and huNRG immunized with iDCgB (Figure 11C). The VAC cohort showed comparable IgG^+^ reactivity against gB as previously observed with the cellular iDCgB vaccine (see also [17]).To summarize, these results showed that upon immunization with VAC, huNRG mice produced detectable IgG responses against HCMV/gB.

## 4. Discussion

HIS-mice are relevant translational and personalized in vivo models and several innovations have been achieved to improve their human adaptive immune reconstitutions. Transgenic mice expressing different combinations of human cytokines, receptors, ligands, and HLAs have significantly contributed to the advances of HIS models. Yet the limitations comprise the efforts and infrastructure needed for generation, testing, breeding, and keeping different transgenic strains simultaneously.

Therefore, here we have considered in vivo lentiviral gene delivery systems to enable the direct transfer of human immunological elements into mice. The practical gain for this approach would be to maintain a single mouse strain and equip it with transgenes empowering human adaptive immunity. Since, there are currently no available in vitro cell culture systems that promote human B cell class switch, one particular emerging application of humanized mice is the generation of fully human monoclonal antibodies (MoAbs). In this regard, we recently demonstrated that engineered iDCgB autologous to the human hematopoiesis in HIS-NRG mice significantly boosted the generation of IgG^+^ B cells for the production of gB-specific MoAbs with antiviral neutralizing activities [17].

Here, we developed a simpler approach based solely on the systemic delivery of lentiviral vectors into humanized mice. We constructed and validated in vitro the tricistronic LV-DR4/fLuc containing the DR promoter driving co-expression of the membrane-bound DR4 dimer and fLuc. Peptides binding to DR4 on 3T3 cells were presented effectively and specifically to DR4-restricted 58-T cells. *HLA-DRB1*0401* was chosen based on the fact that the reference transgenic DRAG mice showed enhanced human B cell development and production of IgGs against pathogens compared with NRG [12,13]. In humans, DR4 was associated with significant protection and clearance of human hepatitis B infections [32]. Although a precise mechanism by which *HLA-DRB1*0401* expression results in high immune reactivity was not fully clarified, studies in transgenic immunocompetent mice challenged with H1N1 Influenza virus suggested that the intracellular trafficking of *HLA-DRB*0401* occurs through the late endosome/lysosomes, resulting in a superior output of innate responses for clearance of virus infections than the non-protective *HLA-DRB*0402* haplotype [33]. Interestingly, DR4 is the strongest known genetic risk factor for rheumatoid arthritis (RA), an autoimmune disease characterized by the abundance of activated B cells and auto-reactive antibodies [34,35]. A paradigm regarding the maintenance of the autoimmune disease-prone DR4 in human populations is that this haplotype can be protective against infections.

In the first step, we evaluated the i.v. LV-DR4/fLuc administration to assess the effects of the systemic DR4 gene delivery in humanized mice. No toxicity or GvHD signs were observed. BLI analyses showed most of the vector expression in the spleen and liver, and persisting long-term. The preferential in vivo transduction of spleen and liver cells is likely to be an effect of the virus retention in these highly fenestrated tissues. Noteworthily, B cells transition from immature to mature cells in the spleen [36], and therefore the HLA class II-mediated antigen presentation in the spleen might provide activation to B cells for class switching.

Then, in the second step, mice were vaccinated with the LV-G2α/gB tricistronic vector to test the immune effects of DR4 gene delivery. This vaccination supplemented the HIS-mice with relevant immune-activating human cytokines and an immunodominant viral antigen known to promote strong IgG responses in humans. DR4/VAC promoted modest effects on the maturation of CD4^+^ T cells, observed as high absolute cell numbers of terminally differentiated CD45RA^−^CD62L^+^ CD4^+^ T cells in the spleen. Concurrently, we observed a significantly higher development of human IgG^+^ and IgA^+^ B cells in the spleen of mice administered with DR4/VAC in comparison with controls. This fits the paradigm that mature human B cells emerging in the spleen of HIS-mice can be primed to recognize foreign antigens and drive their activation, proliferation, hypermutation, and longevity. Nonetheless, under the current experimental conditions, we could only detect gB-specific IgG in mice receiving solely VAC. One possible explanation is that DR4/VAC might have promoted a broader breadth of IgGs, whereby the gB-specific antibodies became relatively diluted. In order to address this assumption, sensitive single B cell IgG repertoire analyses remain to be performed as we recently reported for the iDC/gB vaccine [17]. Incidentally, this work also opened perspectives for further clinical development of the LV-G2α/gB tricistronic vector as a vaccine against HCMV reactivations. In this regard, we will evaluate the effects of VAC or DR4/VAC in controlling or clearing HCMV in our established humanized mouse models [37,38].

Further improvements of the modular LV delivery can be extrapolated, such as applying higher LV doses, evaluating different administration routes, or evaluating vectors with stronger promoters or LVs targeted to infect particular tissues could be evaluated [20]. One promising methodology to be considered is to apply LV-HLA directly intrathymically to improve the HLA-restricted T cell development. Intrathymic LV application after the surgical intervention was shown to be possible [39,40]. Therefore, a rational option could be the combined intrathymic DR4 administration into HIS-mice followed by VAC systemically or subcutaneously.

Finally, one additional interesting future consideration is to test the i.v. LV administrations in NSG-(K^b^-D^b^)^null^ (IA)^null^ (also known as NSG-Dko). Mice homozygous for the five mutations are viable, normal size, and show no gross physical or behavioral abnormalities. This novel strain combines the features of severe combined immunodeficiency with the knock-out of the mouse MHC class I and II molecules. NSG-Dko administered with adoptive T cells shows lower levels of xenogeneic GvHD [41]. This strain combined with the direct administration of LVs could provide a model in which in vivo functionality and mechanisms of T cell responses can be examined and antigen-specific immune therapeutics could be quickly assessed.

Additional research is being carried on with LV expressing the HLA-A* 02:01 haplotype, and other HLAs are in the pipeline. In the long term, we would like to create a toolbox of lentiviral vectors that can be mixed and matched in different combinations in HIS-mice.

## 5. Conclusions

We developed a novel LV-mediated HLA gene delivery to enable a better match of the MHCs of mouse and human cells in HIS-mice. As a “proof of concept”, we tested the HLA-DRB1*04:01 class II haplotype. Mice administered i.v. with the lentivirus expressing HLA-DRB1*04:01 and subsequently immunized with a lentiviral vaccine expressing GM-CSF/IFN-a/HCMV-gB showed significant enhancement in the development of IgG^+^ and IgA^+^ human B cells homing in the spleen. These vector systems can be further developed as new vaccines or to produce human MoAbs in HIS-mice. In addition, the LV-HLA approach can be used to test preclinically in vivo adoptive human T cells restricted to certain HLA-restricted epitopes. In summary, the generation of a lentiviral vector toolbox for in vivo delivery will enable more flexible HLA functionalization of humanized mice.

## 6. Patents

Nothing to disclose.

## Figures and Tables

**Figure 1 biomedicines-09-00961-f001:**
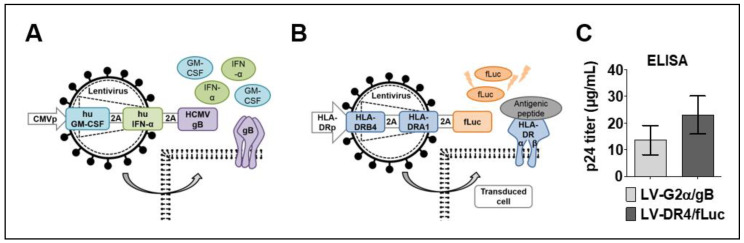
Lentiviral vector designs and viral titer. (**A**) Schematic representation of LV-G2α/gB vaccine (VAC) for Co-expression of secreted human GM-CSF, secreted human IFN-α, and membrane-bound cytomegalovirus glycoprotein B (gB) trimer; (**B**) schematic representation of LV-DR4/fLuc vector for expression of the DR4 dimer on the cell surface and intracellular expression of firefly luciferase (fLuc); and (**C**) average HIV-p24 equivalent viral titers obtained for LV-G2α/gB (n = 5 lots) and LV-DR4/fLuc (n = 7 lots) concentrated vectors.

**Figure 2 biomedicines-09-00961-f002:**
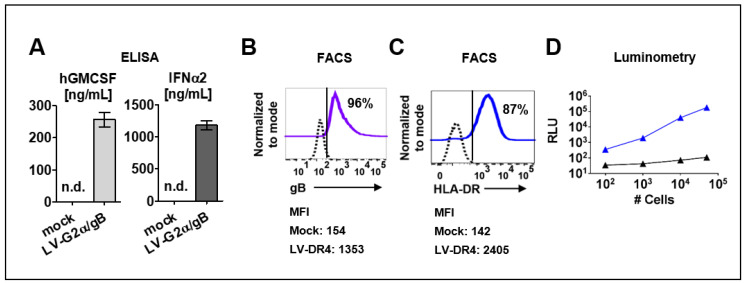
Detection of transgene expression in transduced cell lines. (**A**) Detection of secreted human GM-CSF and human IFN-α in cell supernatants of 293T/ Mock and 293T/ LV-G2α/gB-transduced cells. n.d. = not detected; (**B**) 293T cells transduced with LV-G2α/gB and analyzed by flow cytometry showed expression HCMV/gB on the cell surface. Mock (dashed black line) and LV-G2α/gB transduced (purple line). Representative results from triplicate experiments. The mean fluorescence intensity (MFI) is indicated below; (**C**) 3T3 cells transduced with LV-DR4/fLuc and analyzed by flow cytometry showing expression of the HLA-DR dimer on the cell surface. Mock (dashed black line) and LV-DR4/fLuc transduced (blue line). Representative results from triplicate experiments. The mean fluorescence intensity (MFI) is indicated below; (**D**) analyses of luminescent signals in cell lysates obtained from 3T3 mock cells (black line) and 3T3 cells transduced with LV-DR4/fLuc (blue line).

**Figure 3 biomedicines-09-00961-f003:**
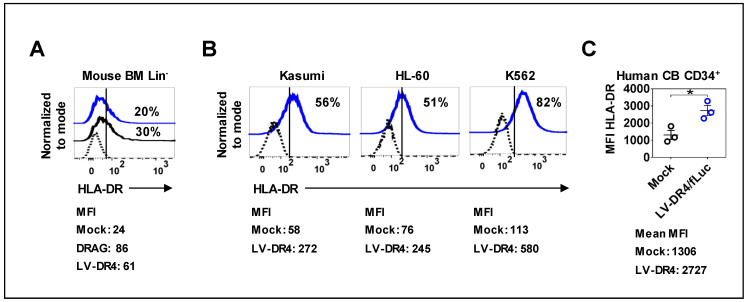
Surface expression of DR4 dimer on hematopoietic cells analyzed by flow cytometry. (**A**) LV-DR4/fLuc transduction of Lin^−^ mouse bone marrow cells obtained from NRG mice and detection of HLA-DR dimer on the cell surface. Lin^−^ mouse bone marrow cells obtained from DRAG mice were used as a positive control for HLA-DR expression analyses (black line). Mock (dashed black line) and LV-DR4/fLuc transduced (blue line). Representative results from duplicate experiments; (**B**) Human Kasumi, HL-60, and K562 leukemia cell lines showing expression of HLA-DR dimer on the cell surface after LV-DR4/fLuc transduction. Mock (dashed black line) and LV-DR4/fLuc transduced (blue line). Representative results from duplicate experiments; (**C**) LV-DR4/fLuc transduction of CB CD34^+^ cells and upregulation of HLA-DR expression (n = 3). Mock (black dots) and LV-DR4/fLuc transduced (blue dots). * *p* < 0.05 (*t*-test, unpaired). The mean fluorescence intensity (MFI) is indicated in each panel.

**Figure 4 biomedicines-09-00961-f004:**
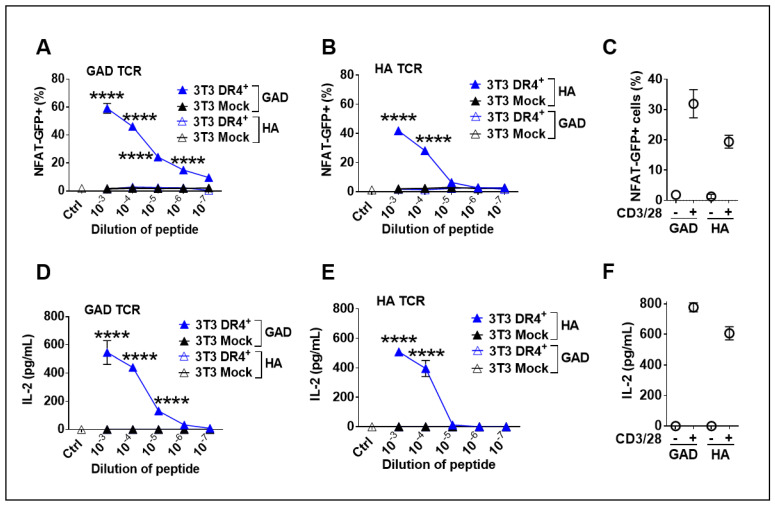
Functional analysis of DR4 dimer on 3T3 cells to present antigens to DR4-restricted 58-T cells leading to their activation. (**A**,**B**) Co-culture of transduced DR4^+^ 3T3 or Mock 3T3 cells with 58 T cell line (expressing GAD or HA TCR together and NFAT-GFP reporter) and stimulation with different dilutions of glutamate decarboxylase (GAD) peptide (**A**) or hemagglutinin (HA) peptide (**B**) or anti-CD3/CD28 (positive control, (**C**) for 24 h. Quantification of glowing GFP^+^ cells was performed by flow cytometry. Secreted IL-2 was measured in supernatant of GAD TCR cells (**D**), HA TCR cells (**E**), and cells treated with anti-CD3/28 (**F**). **** *p* < 0.0001 by Two-way ANOVA with Sidak’s multiple comparison test (n = 3). Data are representative of two independent experiments.

**Figure 5 biomedicines-09-00961-f005:**
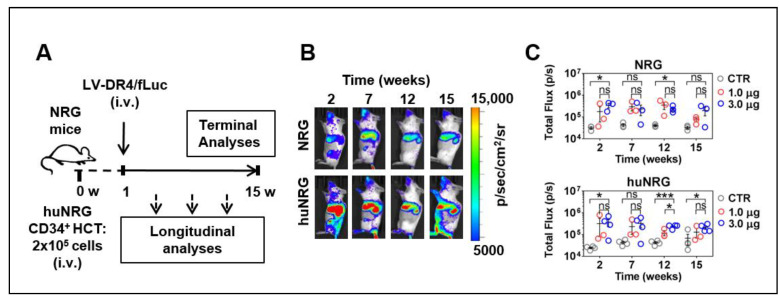
Administration of LV-DR4/fLuc vector intravenously (i.v.) into NRG and huNRG mice and bioluminescence imaging (BLI) analyses. (**A**) Experimental design for testing of LV-DR4/fLuc vector in NRG (CTR: n = 3, 1 μg: n = 3, 3 μg: n = 3), and huNRG mice (CTR: n = 4, 1 μg: n = 3, 3 μg: n = 4); (**B**) Representative BLI pictures of NRG and huNRG mice injected i.v. with 3 μg p24 equivalents of LV-DR4/fLuc vector; (**C**) Quantitative BLI signal obtained by quantifying region of interest (ROI) in spleen from each mouse in NRG and huNRG mice injected i.v. with PBS (CTR), 1 μg, and 3 μg p24 equivalents of LV-DR4/fLuc vector. Dose-finding experiments. * *p* < 0.05, *** *p* < 0.001, ns: not statistically significant (*t*-test, unpaired).

**Figure 6 biomedicines-09-00961-f006:**
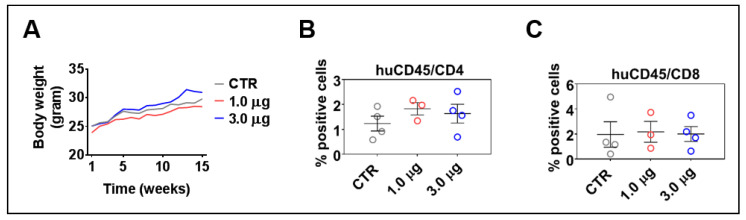
Administration of LV-DR4/fLuc vector into huNRG is safe and shows a trend towards increased CD4^+^ T cell frequencies. (**A**) Weight monitoring of huNRG mice for each cohort after LV-DR4/fLuc administration; (**B**,**C**) terminal analyses of the frequencies of CD4^+^ and CD8^+^ T cells in the spleen by flow cytometry. No significant effects were observed in these dose-finding experiments.

**Figure 7 biomedicines-09-00961-f007:**
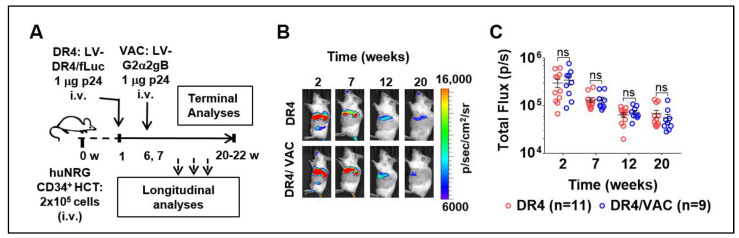
Administration of LV-DR4/fLuc vector (DR4, n = 11) and in combination with LV-G2α/gB (DR4/VAC, n = 9) intravenously (i.v.) into huNRG mice and bioluminescence imaging (BLI) analyses. (**A**) Experimental design for testing the combined effect of DR4 and VAC in huNRG mice; (**B**) representative BLI picture of huNRG mice injected i.v. with 1 μg p24 equivalents of LV-DR4/fLuc vector alone (DR4) or in a combination of 1 μg p24 equivalents of LV-G2α/gB (DR4/VAC); (**C**) quantitative BLI signal obtained by quantifying ROI in spleen from each mouse in DR4 and DR4/VAC cohorts. Data pooled from four independent experiments. ns: not statistically significant; Statistical analyses were performed using the Welch *t*-test applied to log data (unpaired).

**Figure 8 biomedicines-09-00961-f008:**
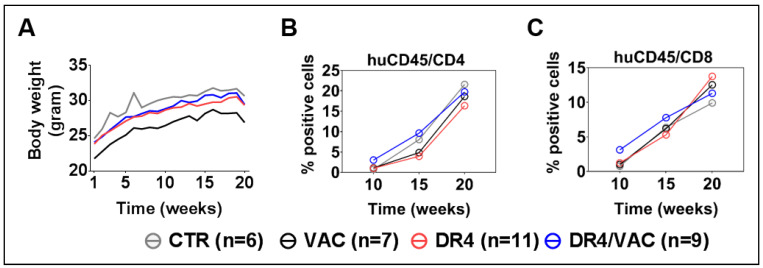
Administration of LV-DR4/fLuc vector in combination with VAC into huNRG is safe and promotes CD4^+^ and CD8^+^ T cell development and maturation. (**A**) Weight monitoring of huNRG mice for each cohort after HCT; (**B**,**C**) longitudinal analyses of CD4^+^ and CD8^+^ T cells frequency in peripheral blood by FACS. Data pooled from four independent experiments.

**Figure 9 biomedicines-09-00961-f009:**
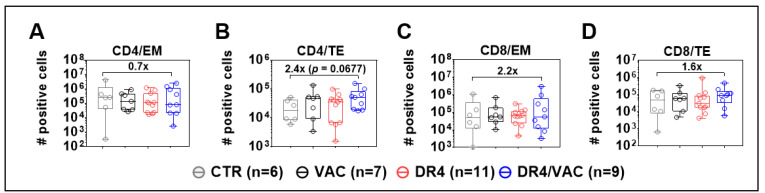
DR4 combined with VAC promotes CD4^+^ and CD8^+^ T cell maturation. Terminal analyses of absolute numbers (#) of positive cells identified as CD4^+^ effector memory (EM) (**A**); CD4^+^ terminal effector (TE) (**B**); CD8^+^ effector memory (EM) (**C**); and CD8^+^ terminal effector (TE) (**D**) in spleen by FACS. X-fold difference of the mean comparing the values obtained for controls and DR4/VAC mice is indicated in the plots. Data pooled from four independent experiments. Statistical analyses were performed using the Welch *t*-test applied to log data (unpaired).

**Figure 10 biomedicines-09-00961-f010:**
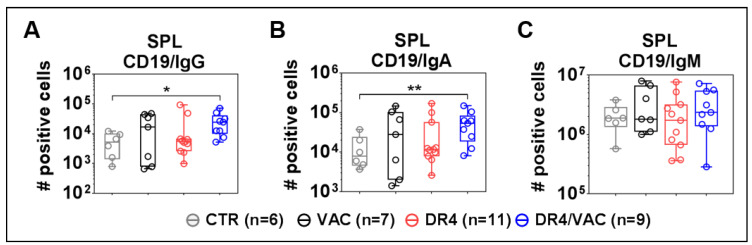
VAC and DR4/VAC promotes B cell maturation and response. (**A**–**C**) Terminal analyses of of absolute numbers (#) of positive cells identified as CD19^+^IgM^−^IgG^+^, CD19^+^IgM^−^IgA^+^, and CD19^+^IgM^+^ B cell subsets in the spleen by flow cytometry. Data pooled from four independent experiments. * *p* < 0.05, ** *p* < 0.01. Statistical analyses were performed using the Welch *t*-test applied to log data with Laplace correction (+1) (unpaired).

**Figure 11 biomedicines-09-00961-f011:**
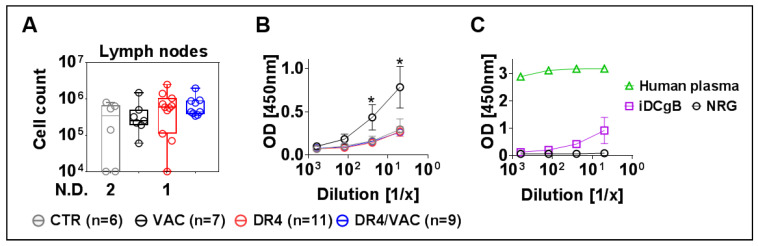
DR4/VAC promotes LN cellularity and VAC results in anti-gB IgG^+^ responses. (**A**) Total cell count in lymph nodes. No detection (N.D.) refers to the number of mice; (**B**,**C**) detection of IgG reactivity against gB in plasma by ELISA. * *p* < 0.05 when comparing DR4 vs. VAC. Statistical analyses were performed using the Welch *t*-test applied to log data.

## Data Availability

The descriptive statistical analyses data supporting reported results can be found in Appendix A. Additional data are available upon request.

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
