# Peer review of "In Vivo Lentiviral Gene Delivery of HLA-DR and Vaccination of Humanized Mice for Improving the Human T and B Cell Immune Reconstitution"

_biomedicines, 2021, doi:10.3390/biomedicines9080961_

Round 1

Reviewer 1 Report

In this manuscript, Kumar et al. evaluated in vivo co-delivery of lentiviral vectors (LVs) for expression of a human leukocyte antigen (HLA-DRA*01/ HLA-DRB1*0401, DR4), and a LV vaccine expressing human cytokines (GM-CSF and IFN-a) and a human cytomegalovirus gB antigen 38 (HCMV-gB). The authors have shown that  i.v. injection with LV-DR4/fLuc 39 led to long-lasting vector expression in multiple organs in humanized NRG mice. Furthermore, administration of the LV vaccine after LV-DR4/fLuc enhanced the cellularity of lymph nodes, maturation of terminal effector CD4 T cells, and promoted significantly higher development  of IgG+ and IgA+ B cells.

Overall, the authors have provided convincing evidence to support the future use of in vivo systemic LV-mediated HLA-DR delivery and vaccination to promote development of human class-switched B cells in HIS-mice. I have a couple of minor points that could further strengthen the manuscript.

  1. Figure 2A should include controls for the ELISA.
  2. In Figure 3A-C, MFI of HLA-DR should be added.

Author Response

  1. “Figure 2A should include controls for the ELISA”: the mock control was added. We also included the MFI in Fig. 2B as it was requested for Fig. 3 (see below). The legend was revised and the text is highlighted.
  2. “In Figure 3A-C, MFI of HLA-DR should be added”: The MFIs were included in Fig. 3A-C. The revisions of the legend and text are highlighted. The legend was revised and the text is highlighted.

Reviewer 2 Report

The manuscript entitled “In vivo lentiviral gene delivery of HLA-DR and vaccination into humanized mice for improving the human T and B cell immune reconstitution” describe the new developed LV-mediated HLA gene delivery to enable a better match of the MHCs of mouse and human cells in HIS-mice. As a “proof of concept” we tested the HLA-DRB1*04:01 class II haplotype. Mice administered i.v. with the lentivirus expressing HLA-DRB1*04:01 and subsequently immunized with a lentiviral vaccine expressing GM-CSF/ IFN-a/ HCMV-gB showed significant enhancement in the development of IgG+ and IgA+ human B cells homing in the spleen. These vector systems can be later futher develop as new vaccines or to produce human MoAbs in HIS-mice. In addition, the LV-HLA approach can be used to test preclinically in vivo adoptive human T cells restricted to certain HLA-restricted epitopes. In summary, the generation of a lentiviral vector toolbox for in vivo delivery will enable more flexible HLA functionalization of humanized mice.

English language and style check are required.

Author Response

“English language and style check are required”. The text was extensively edited. Changes are highlighted.